# Conformal Prediction for Uncertainty-Aware Planning with Diffusion Dynamics Model

**Jiankai Sun**
Stanford University
jksun@stanford.edu

**Yiqi Jiang**
Stanford University
yqjiang@stanford.edu

**Jianing Qiu**
Imperial College London
jianing.qiu17@imperial.ac.uk

**Parth Talpur Nobel**
Stanford University
ptnobel@stanford.edu

**Mykel Kochenderfer**
Stanford University
mykel@stanford.edu

**Mac Schwager**
Stanford University
schwager@stanford.edu

## Abstract

Robotic applications often involve working in environments that are uncertain, dynamic, and partially observable. Recently, diffusion models have been proposed for learning trajectory prediction models trained from expert demonstrations, which can be used for planning in robot tasks. Such models have demonstrated a strong ability to overcome challenges such as multi-modal action distributions, high-dimensional output spaces, and training instability. It is crucial to quantify the uncertainty of these dynamics models when using them for planning. In this paper, we quantify the uncertainty of diffusion dynamics models using Conformal Prediction (CP). Given a finite number of exchangeable expert trajectory examples (called the "calibration set"), we use CP to obtain a set in the trajectory space (called the "coverage region") that is guaranteed to contain the output of the diffusion model with a user-defined probability (called the "coverage level"). In PlanCP, inspired by concepts from conformal prediction, we modify the loss function for training the diffusion model to include a quantile term to encourage more robust performance across the variety of training examples. At test time, we then calibrate PlanCP with a conformal prediction process to obtain coverage sets for the trajectory prediction with guaranteed coverage level. We evaluate our algorithm on various planning tasks and model-based offline reinforcement learning tasks and show that it reduces the uncertainty of the learned trajectory prediction model. As a by-product, our algorithm PlanCP outperforms prior algorithms on existing offline RL benchmarks and challenging continuous planning tasks. Our method can be combined with most model-based planning approaches to produce uncertainty estimates of the closed-loop system.

## 1 Introduction

Uncertainty is a major challenge in robotics, arising from a variety of sources, including errors in modeling, sensing, planning, and control [1, 2, 3]. Understanding the effect of uncertainty is important for developing successful motion strategies, as it allows robots to anticipate and plan for unexpected events. Therefore, planning with uncertainty is a crucial challenge in robotics and accurately modeling this uncertainty is important for ensuring the safety and reliability of robotic systems, particularly in safety-critical applications. A range of techniques, such as probabilistic modeling [4] and robust optimization [5, 6], can be used to account for the uncertainty of dynamics models in robotics.

37th Conference on Neural Information Processing Systems (NeurIPS 2023).

The dynamics model is especially critical for model-based planning [7] and reinforcement learning (MBRL) [8]. Due to the advantages such as expressing multi-modal action distributions, high-dimensional output space, and stable training, there has been a growing trend in learning-based planning approaches to utilizing diffusion models as dynamics models [9, 10]. Diffusion models can also address the challenge of multi-modal distribution of actions in planning, which arises when multiple actions may be appropriate for a given state. This is achieved by enabling policies to express a distribution over multiple actions rather than a deterministic choice of a single action [11]. Furthermore, diffusion models hold the potential to be applied to Offline Reinforcement Learning, which aims to learn effective strategies using offline datasets without exploration [12]. However, existing model-based algorithms struggle when encountering out-of-distribution (OOD) actions or states [9]. A crucial element absent from these methods is the adequate management of uncertainty in the offline scenario [13, 14]. Uncertainty-aware dynamics models that convey algorithmic confidence are valuable if they possess accurate calibration and sharpness [15]. Calibration entails aligning predicted probabilities with actual outcomes, whereas sharpness refers to the concentration of predicted probabilities around the true probability [16]. Nevertheless, many machine learning algorithms generate poorly calibrated probability estimates, which can result in overconfident or underconfident predictions and lead to suboptimal decision-making [17, 18]. Therefore, to ensure accuracy and optimal decision-making, it is crucial to utilize uncertainty-aware models that are well-calibrated and sharp.

Conformal Prediction (CP) [19] is a potent approach to assessing the uncertainty associated with predictions generated by black-box models. CP offers several advantages over traditional methods for measuring uncertainty. For example, CP is model-agnostic [20], meaning it can be applied to any type of model without requiring knowledge of the model's inner mechanisms. This is particularly advantageous in cases where the underlying model is complex and not well understood. Additionally, CP allows for customization of the desired confidence level, providing the ability to control the trade-off between predictive accuracy and the level of uncertainty [21]. This feature is especially important in safety-critical applications where a higher level of confidence is required. Furthermore, CP can be used for online learning [22], meaning that it can adapt to new data points as they become available. This makes it a particularly useful approach for planning in dynamic environments. Lastly, CP is distribution-free [23], providing statistical guarantees [24] and reliable uncertainty estimates from finite samples [25]. Given these advantages, we choose to use CP as our method of uncertainty quantification for diffusion dynamics models in planning. The flexibility and reliability of CP provide a rigorous and mathematically proven framework for uncertainty quantification, which is critical for ensuring the safety and reliability of robotic systems in a variety of applications.

In this paper, we connect conformal prediction to planning and propose `PlanCP`, adopting a practical and effective conformal prediction uncertainty estimation method for planning algorithms. Empirically, we observe that `PlanCP` substantially improves model stability. In addition, `PlanCP` reduces uncertainty without sacrificing performance on datasets with demonstrations collected from experts. `PlanCP` can be applied to reinforcement learning or imitation learning from demonstrations [26, 27] as well. However, since we build on the Diffuser [9], which calls itself a planning method, we will generally stick with the planning terminology for consistency with the literature.

In summary, our main contributions are as follows:

- We introduce `PlanCP`, a framework that enables the use of a conformal prediction to estimate uncertainty in planning tasks, without sacrificing the accuracy of dynamics models.
- Our approach is compatible with diffusion dynamics models. We demonstrate that by employing a diffusion model as the foundational dynamics model, we can construct a conformal predictor that guarantees trustworthy and accurate uncertainty estimates.
- We extensively evaluate our approach, and the experimental results demonstrate that it exhibits lower uncertainty without compromising performance across multiple benchmarks, including Offline Reinforcement Learning.

## 2 Related Work

### 2.1 Conformal Prediction

Conformal Prediction (CP) [19, 28, 29, 30, 21, 24, 31, 32, 33] is effective for measuring the uncertainty associated with predictions produced by black-box models. As a variant of conformal

prediction, Transductive Conformal Predictor (TCP) can be applied on top of existing machine learning methods to establish valid prediction regions, [19]. TCP takes into account the structure of the data being predicted and uses the entire dataset during the prediction phase to construct prediction regions around each data point [34]. To improve the computational efficiency of the initial transductive approach, a more computationally efficient framework Inductive Conformal Predictor (ICP) was proposed, which involves dividing the training data into a proper training set and a calibration set [35]. However, ICP can be informationally inefficient as some examples are used solely for modeling while others are used only for calibration. To address this, ensembles of conformal predictors have been proposed [36, 37] and [38]. Examples include Cross Conformal Predictor (CCP) [39], Bootstrap Conformal Predictor (BCP) [19] and the generalized Aggregated Conformal Predictor (ACP) [40]. In a recent study [41], conformal prediction was used to construct prediction regions for recurrent neural networks (RNNs). Strawn et al. [42] propose conformal predictive safety filters that use statistical techniques to provide uncertainty intervals around predictions and learn an additional safety filter that closely follows the RL controller but avoids uncertainty intervals. Tonkens et al. [43] expand upon prior work in planning with probabilistic safety guarantees using conformal prediction. Conceptually closest to our work are [44, 45, 42, 43]. However, in these works, only quantification of uncertainty is provided, and reduction of the uncertainty during training is absent. In comparison, we are able to reduce uncertainty in training and extend conformal prediction to diffusion models.

## 2.2 Diffusion Models

In recent years, diffusion models have gained significant attention as a powerful class of generative models that formulates the data-generating process as a sequence of denoising steps [46, 47, 48, 11]. Inspired by score matching [49] and energy-based models (EBMs) [50, 51, 52, 53], diffusion models rely on a denoising procedure, which can be viewed as a way of parameterizing the gradients of the data distribution [54]. From the model-based trajectory-planning perspective, Diffuser [9] applies a diffusion model as a trajectory generator. From the model-free policy-optimization perspective, Wang et al. [11] utilize a diffusion model to represent the policy, applying it to the action space and conditioning it on the states to form a conditional diffusion model. Diffusion Policy [10] utilizes a conditional diffusion-based process to model policies for real robots, exhibiting robust handling of multi-modal action distributions. However, the aforementioned approaches lack the capability to quantify the associated uncertainty, which is precisely the focus of our paper.

## 2.3 Planning, Predictive Dynamics Model, and Model-Based Reinforcement Learning

Model-based reinforcement learning (MBRL) has emerged as a promising solution for real-world sequential decision-making problems with distinct advantages such as analytic gradient computation [55]. A significant body of literature focuses on learning a dynamics model and utilizing the learned model for policy learning through model-based planning [56, 57, 58, 59]. PILCO [60] utilizes a Gaussian process to model system dynamics, while World Models [61] and Dreamer [62] learn latent dynamics through two-stage and online planning, respectively. MGAIL [63] introduces a forward model in GAIL, but its discriminators output a uniform representation that is less transferable. PETS [64] combines probabilistic neural networks and deterministic neural networks to create uncertainty-aware models. To capture long-term dependencies, FNNs, RNNs, and Transformers have been used to develop internal models [65, 66, 67]. Our study distinguishes itself from prior research by investigating techniques to measure and minimize uncertainty using conformal prediction for diffusion-based dynamics models.

## 3 Method

In this section, we describe our framework `PlanCP` (as Fig. 1 shows) to solve the problem of uncertainty-aware planning. We begin by outlining our overall methodology, which involves three key steps. First, we introduce the diffusion-based dynamics model to predict future trajectories. Second, we apply the conformal prediction technique on the dynamics model to quantify and reduce the uncertainty. Finally, we summarize the whole procedure of our algorithm. In the following subsections, we provide more details on each of these steps and explain the specific techniques we used to implement them. As for notation, we use superscript $n$ to denote diffusion timestep, $i$ or $k$ to denote different trajectories, and $t$ to denote trajectory timestep.

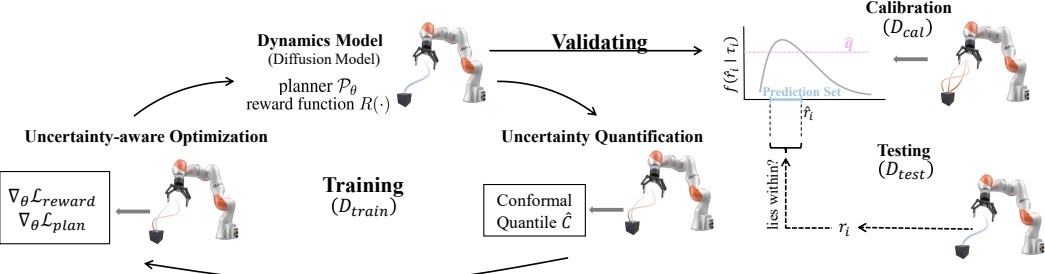

Figure 1: `PlanCP` **Framework:** To update the dynamics model during training, we perform uncertainty-aware optimization using the training set $D_{train}$. During calibration, we construct the uncertainty interval $\hat{q}$ based on $D_{cal}$, and then we verify the uncertainty interval using the test set $D_{test}$.

## 3.1 Background

We consider a Markov Decision Process (MDP) represented by the tuple $(\mathcal{S}, \mathcal{A}, \mathcal{T}, r, \rho_0, \gamma)$ with state space $\mathcal{S}$, action-space $\mathcal{A}$, dynamics $\mathcal{T}(s'|a, s)$, reward function $r(s, a)$, initial state distribution $\rho_0(s)$, and discount factor $\gamma \in (0, 1)$. We assume a stochastic policy $\pi$ that maps a state to a distribution over actions. We define $\tau$ as trajectories, which consist of sequences of state-action pairs $(s_0, a_0, \cdots, s_T, a_T)$ generated by a policy $\pi$. $\tau^E$ are expert trajectories generated by an expert policy $\pi^E$. $T$ represents the time horizon. We use a two-dimensional array to represent the trajectory $\tau$:

$$\tau = \begin{bmatrix} s_0 & s_1 & \dots & s_T \\ a_0 & a_1 & \dots & a_T \end{bmatrix}, \tag{1}$$

with each column corresponding to a timestep in the planning horizon.

### 3.1.1 Planning with Conditional Diffusion Probabilistic Models

In Bayesian inference, it is intractable to compute the exact posterior distribution due to the high dimensionality and the complexity of the model. One type of latent variable model approximates posterior by the forward or diffusion process $q(\tau^n|\tau^{n-1})$, which is modeled as a fixed Markov chain that progressively adds Gaussian noise to the data. Diffusion Probabilistic Models, as described in [46, 47], characterize the process of data generation as an iterative denoising procedure, formalized as the probability distribution $p_\theta(\tau^{n-1}|\tau^n)$. The denoising operation (as Eq. 2 shows) is performed in reverse of a forward process $q(\tau^n|\tau^{n-1})$, which progressively introduces noise and alters the underlying structure of the data.

$$p_\theta(\tau^0) = \int p(\tau^N) \prod_{n=1}^{N} p_\theta(\tau^{n-1}|\tau^n) d\tau_{1:N}, \tag{2}$$

where $p(\tau^N) = \mathcal{N}(\tau^N; \mathbf{0}, \mathbf{I})$ is a standard Gaussian prior and $\tau^0 \sim q(\tau^0)$ is noiseless problem data. The reverse process is parameterized as Gaussian (Eq. 3):

$$p_\theta(\tau^{n-1}|\tau^n) = \mathcal{N}(\tau^{n-1}; \mu_\theta(\tau^n, n), \mathbf{\Sigma}_n), \tag{3}$$

in which $\mu_\theta$ and $\Sigma$ are the mean and covariance of the Gaussian distribution respectively. Janner et al. [9] proposed a way to turn diffusion models into planners that can incorporate different conditions. To achieve this, a learned/expert-designed condition function $w(\cdot)$ is added to the diffusion model, which yields a planner $\mathcal{P}_\theta$ described by Eq. 4:

$$\mathcal{P}_\theta(\tau) \propto p_\theta(\tau)w(\cdot). \tag{4}$$

where $p_\theta(\tau)$ is the diffusion model's probability distribution, and $w(\cdot)$ is a conditional function that can include information such as observations history, desired goals, or rewards. Practically, this can be implemented by sampling from the unperturbed reverse process and replacing the sampled values with conditioning values $w(\cdot)$ at all diffusion timesteps. We can use a learned reward function as a condition, as introduced in Sec. 3.2.

**Algorithm 1:** `PlanCP`: Conformal Prediction for Planning with Diffusion Dynamics Models

---

**Data:** Training set $D_{train}$ with $K_{train}$ samples, calibration set $D_{cal}$ with $K_{cal}$ samples, test set $D_{test}$ with $K_{test}$ samples, planner $\mathcal{P}_\theta$, expert trajectory $\boldsymbol{\tau}^E$, and failure probability $\alpha$.

**Result:** predicted trajectory $\boldsymbol{\tau}$, uncertainty $C$.

1 **while** *not converged* **do**
2     **for** $i = 0, 1, 2, \cdots, K_{train} - 1$ **do**
3         Get predicted trajectory $\boldsymbol{\tau}_i$ corresponding to expert trajectory $\boldsymbol{\tau}_i^E \in D_{train}$;
4         Compute *reconstruction nonconformity score* as Eq. 6 for each $\boldsymbol{\tau}_i^E \in D_{train}$;
5         Construct uncertainty $C_i$ as Eq. 9;
6         Compute total loss $\mathcal{L}_{plan}$ as Eq. 13 and $\mathcal{L}_{reward}$ as Eq. 14;
7         Update the dynamics model $\mathcal{P}_\theta$ using loss $\mathcal{L}_{plan}$ and the reward model $R(\cdot)$ using loss $\mathcal{L}_{reward}$;
8     **end**
9 **end**
10 **for** $i = 0, 1, 2, \cdots, K_{cal} - 1$ **do**
11     Get predicted trajectory $\boldsymbol{\tau}_i$ as $D_{cal}$;
12     Compute *reward conformity score* $r_i$ for each $\boldsymbol{\tau}_i \in D_{cal}$;
13     Construct predicted interval $\hat{q}$ as Eq. 15;
14 **end**
15 Verify the predicted interval using the test set $D_{test}$;

---

## 3.2 Learned Reward Function

Learning a reward model is important because it allows an intelligent agent to estimate the expected rewards of a given action and state, which is useful for the agent to make informed decisions in an environment. Additionally, using a learned reward model can enable us to adapt to changes in the environment and adjust our predictions accordingly. However, learning an accurate reward model from data is challenging. Thus, we introduce a transformer-based reward model as Eq. 5, which can perform accurate long-term temporal predictions.

$$
\begin{aligned}
\hat{r}_{0:T} &= R(\text{MultiHead}(Q = \boldsymbol{s}_{0:T}, K = \boldsymbol{s}_{0:T}, V = \boldsymbol{a}_{0:T})), \\
\text{MultiHead}(Q, K, V) &= \text{Concat}(\text{head}_1, \text{head}_2, ..., \text{head}_h)W^O, \\
\text{head}_j &= \text{Attention}(QW_j^Q, KW_j^K, VW_j^V), \\
\text{Attention}(Q, K, V) &= \text{softmax}\left(\frac{QK^T}{\sqrt{d_f}}\right) V,
\end{aligned}
\tag{5}
$$

where $R(\cdot)$ is an MLP layer to map the multi-head attention feature to a predicted reward. $d_f$ is the dimension of the attention feature. For more details of transformer architecture, please refer to [68]. The transformer-based reward model is used as a condition for diffusion models to generate reasonable trajectories.

## 3.3 Quantifying Uncertainty with Conformal Prediction

The inherent uncertainty in the diffusion-based dynamics model $\mathcal{P}_\theta$ can lead to inaccurate predictions. To address these challenges, we employ the conformal prediction technique to obtain prediction uncertainty. Unlike other approaches [4], conformal prediction does not rely on assumptions about the underlying distribution or the predictive model, making it a flexible and robust method for uncertainty quantification in machine learning [24]. This allows us to quantify the uncertainty associated with the predictions and provides more robust estimates of the model's performance.

We have split the dataset $D$ into three parts — $D_{train}$, $D_{cal}$, and $D_{test}$ — for training, calibration, and testing, respectively. The planner $\mathcal{P}_\theta$ and reward function $R(\cdot)$ are learned from the $D_{train}$ dataset.

In our problem setting, we use *reconstruction nonconformity scores* during training and *reward conformity scores* during calibration and testing. *Reconstruction nonconformity scores* is defined as

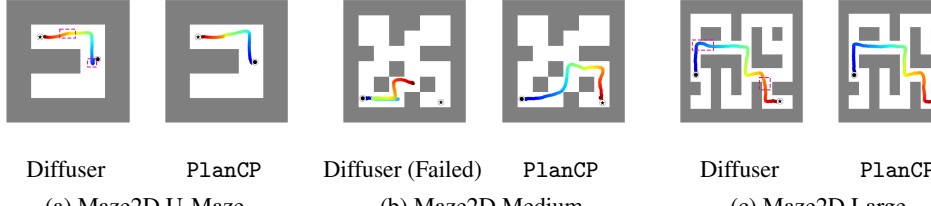



Diffuser     PlanCP     Diffuser (Failed)     PlanCP     Diffuser     PlanCP

(a) Maze2D U-Maze     (b) Maze2D Medium     (c) Maze2D Large



Figure 2: **Learned long-horizon planning**: By learning the planning procedure, `PlanCP` is capable of conducting long-horizon planning with a specified start ◉ and goal ✪ condition. even under conditions of sparse rewards. Sub-environment visualization for Maze2D: (a) U-Maze, (b) Medium, (c) Large. We have marked the risky region in the baseline with purple dashed boxes.

the mean squared error of the learned trajectories from the expert trajectories. *Reward conformity scores* are exactly the reward returned by the environment for a given trajectory.

**Training** To be able to compute and reduce the uncertainty during training, we need to perform conformal prediction on the training set $D_{train}$. First, we compute the *reconstruction nonconformity score* as Eq. 6 for each trajectory $\tau_i$ of the training set $D_{train}$. A lower *reconstruction nonconformity score* corresponds to better predictions.

$$\tilde{r}_i = ||\tau_i - \tau_i^E||_2, \tag{6}$$

where $\tau_i$ is the predicted $i$th trajectory while $\tau_i^E$ is the ground-truth $i$th trajectory from the expert demonstration of $D_{train}$.

**Calibration and Testing** During calibration and testing, we use *reward conformity score* to quantify the uncertainty. Let $r_0, r_1, \cdots, r_k$ be $k+1$ rewards corresponding to $\tau_0, \tau_1, \cdots, \tau_k$ trajectories predicted by our planner $\mathcal{P}_\theta$. These rewards are defined and returned by the environments, which can be regarded as independent and identically distributed random variables. For planning, the *reward conformity score* can be seen as a metric that reflects the planner's ability to predict the quality of plans. A higher *reward conformity score* corresponds to better predictions, indicating that the planner is more accurate in its estimations. Our objective is to quantify the uncertainty associated with the reward $r_i$ based on a set of reference rewards $\{r_0, \cdots, r_k\} \setminus r_i$. Specifically, we aim to construct a prediction interval $C_i$ that contains $r_i$ with probability at least $1 - \alpha \in (0, 1)$. Formally, we seek to find a valid prediction interval $C_i$ that satisfies the following inequality:

$$\mathbb{P}(r_i \in C_i(\{r_0, \cdots, r_k\} \setminus r_i)) \geq 1 - \alpha, \tag{7}$$

where $C_i(\{r_0, \cdots, r_k\} \setminus r_i)$ denotes the prediction interval constructed using the reference rewards $\{r_0, \cdots, r_k\} \setminus r_i$. The prediction interval $C_i$ should provide a measure of the uncertainty associated with the reward $r_i$, and it should be constructed in such a way as to guarantee that with a probability of at least $1 - \alpha$, it contains $r_i$.

**Lemma 3.1.** *Given $k+1$ exchangable random trajectories $(\tau_0^E, \tau_1^E, \cdots, \tau_k^E) \sim D_{train}$, predicted trajectory $(\tau_0, \tau_1, \cdots, \tau_k)$, corresponding (non)conformity scores $d_0, d_1, \ldots, d_k$, and a failure probability $\alpha \in (0, 1)$, then*

$$\mathbb{P}[d_i \leq C_i] \geq 1 - \alpha, \tag{8}$$

*where the prediction interval $C_i$ are defined by*

$$C_i = Quantile\left(d_0, \ldots, d_{i-1}, d_{i+1}, \ldots, d_k; \frac{\lceil (k+1)(1-\alpha) \rceil}{k}\right). \tag{9}$$

During training, we use *reconstruction nonconformity score* as $d_i$, while during calibration and testing, we use the *reward conformity score* as $d_i$. Note that if $\alpha$ is less than $1/k$, our confidence intervals become infinitely wide as we have insufficient data to make predictions with that accuracy.

Table 1: **Long-horizon planning in Maze2D.** Maze2D features a sparse reward structure and requires long-horizon planning. `PlanCP` has lower uncertainty and better performance. Bold font means better performance.

| Environment | BCQ [70] | CQL [71] | IQL [72] | Diffuser [9] | PlanCP |
|---|---|---|---|---|---|
| Maze2D U-Maze | 12.8 | 5.7 | 47.4 | $113.9 \pm 3.1$ | $\mathbf{116.4 \pm 3.2}$ |
| Maze2D Medium | 8.3 | 5.0 | 34.9 | $121.5 \pm 2.7$ | $\mathbf{128.5 \pm 2.5}$ |
| Maze2D Large | 6.2 | 12.5 | 58.6 | $123.0 \pm 6.4$ | $\mathbf{130.9 \pm 5.6}$ |
| **Average** | 9.1 | 7.7 | 47.0 | 119.5 | **125.3** |

## 3.4 Trajectory Optimization on Uncertainty-aware Planning

Trajectory optimization for a discrete-time dynamics system defined by $s_{t+1} = f(s_t, a_t)$ involves identifying a sequence of actions $a_{0:T}$ that maximizes an objective function $\mathcal{J}$, which is the sum of per time-step rewards or costs $c(s_t, a_t)$. The optimization problem can be represented as follows:

$$a_{0:T} = \arg\min_{a_{0:T}} \mathcal{J}(s_0, a_{0:T}) = \arg\min_{a_{0:T}} \sum_{t=0}^{T} \gamma^t c(s_t, a_t). \tag{10}$$

Here, $T$ represents the planning horizon.

While training, we utilize the uncertainty calculated from $\{\tilde{r}_i\}$ as a cost function to optimize the parameters of the planner model. After computing the *reconstruction nonconformity scores* on our dataset and setting $\hat{C} = \text{Quantile}\left(\tilde{r}_1, \ldots, \tilde{r}_k; \frac{\lceil (k+1)(1-\alpha) \rceil}{k}\right)$, we can form the uncertainty loss:

$$\mathcal{L}(\theta)_{uncertainty} = \hat{C}. \tag{11}$$

After adopting the differentiable ranking and sorting techniques [69], the soft quantile function is differentiable. On the other hand, a diffusion probabilistic model is used to parameterize the learned gradient $\epsilon_\theta(\tau^n, n)$ of the trajectory denoising process. The objective for training the $\epsilon$ model can be expressed as Eq. 12:

$$\mathcal{L}(\theta)_{recon} = \mathbb{E}_{n, \epsilon, \tau^0}[\|\epsilon - \epsilon_\theta(\tau^n, n)\|_2], \tag{12}$$

where $n \sim \mathcal{U}\{1, 2, \ldots, N\}$ represents the diffusion timestep. $\epsilon \sim \mathcal{N}(0, \mathbf{I})$ represents the noise target. Finally, $\tau^n$ denotes the trajectory $\tau^0$ that is corrupted with noise $\epsilon$.

In the context of learning, we hope to use conformal prediction to construct prediction intervals for expected cumulative rewards, which can be used to estimate the uncertainty of the predictions. By incorporating these prediction intervals into the decision-making process, we can make more informed decisions and avoid overly optimistic or pessimistic estimations. Finally, the learning objective for planner training is

$$\mathcal{L}_{plan} = \mathcal{L}(\theta)_{recon} + \lambda_{uncertainty}\mathcal{L}(\theta)_{uncertainty}. \tag{13}$$

Given predicted reward $\hat{r}$ and ground-truth reward $r$, the learning objective for reward model training is

$$\mathcal{L}_{reward} = \|\hat{r} - r\|_2. \tag{14}$$

The reward model $R(\cdot)$ is used as the condition for the diffusion model as Eq. 4 shows. We can obtain reward $r$ returned by the environment or learned reward model. On the calibration dataset $D_{cal}$, we compute the *reward conformity scores* and set $Q_i^\alpha, Q_i^{1-\alpha}$ and prediction interval $\hat{q}$ using conformal prediction as Eq. 15.

$$Q^\alpha(r_{k+1}) = \text{Quantile}\left(r_1, \ldots, r_k; \frac{\lceil (k+1)(1-\alpha) \rceil}{k}\right),$$

$$Q^{1-\alpha}(r_{k+1}) = \text{Quantile}\left(r_1, \ldots, r_k; \frac{\lceil (k+1)\alpha \rceil}{k}\right), \tag{15}$$

$$\hat{q}(r_{k+1}) = [Q^{1-\alpha}(r_{k+1}), Q^\alpha(r_{k+1})].$$

Once the prediction interval is constructed, we assess its performance on the test set $D_{test}$.

Table 2: **Uncertainty.** The uncertainty quantification of `PlanCP` and prior algorithms in the Maze2D environment. `PlanCP` has lower uncertainty than previous methods for most tasks. Bold font means better performance.

| Dataset | Diffuser [9] | | | PlanCP | | |
|---|---|---|---|---|---|---|
| | Rewards ↑ | Uncertainty ↓ | Interval $\hat{q}$ ↓ | Rewards ↑ | Uncertainty ↓ | Interval $\hat{q}$ ↓ |
| Maze2D U-Maze | $113.9 \pm \mathbf{3.1}$ | [121.83, 147.19] | 25.36 | $\mathbf{116.4} \pm 3.2$ | [121.11, 142.85] | **21.74** |
| Maze2D Medium | $121.5 \pm 2.7$ | [132.77, 141.85] | **9.08** | $\mathbf{128.5} \pm \mathbf{2.5}$ | [132.77, 142.99] | 10.22 |
| Maze2D Large | $123.0 \pm 6.4$ | [92.11, 163.23] | 71.12 | $\mathbf{130.9} \pm \mathbf{5.6}$ | [97.75, 168.46] | **70.71** |
| **Average** | 119.5 | | | **125.3** | | |

## 3.5 Our Proposed Procedures

We first iterate over the training set $D_{train}$ and for each expert trajectory $\boldsymbol{\tau}_i^E$, obtain a predicted trajectory $\boldsymbol{\tau}_i$ using the learned dynamics model $\mathcal{P}_\theta$. Then we compute the *reconstruction nonconformity score* $\tilde{r}$ for each $\boldsymbol{\tau}_i^E$ and construct the uncertainty $C_i$. The loss $\mathcal{L}_{plan}$ and $\mathcal{L}_{reward}$ are then computed using Eq.13 and Eq.14, respectively. Finally, the dynamics model $\mathcal{P}_\theta$ and reward model $R(\cdot)$ are updated using the computed loss.

Next, we iterate over the calibration set $D_{cal}$, and for each predicted trajectory $\boldsymbol{\tau}_i$, obtain the *reward conformity score* $r_i$ and construct the predicted interval $\hat{q}_i$.

Finally, we verify the predicted interval using the test set $D_{test}$ to evaluate its performance. The whole procedure is shown in Alg. 1.

# 4 Experiments

In our experiments, we aim to answer the following questions: (1) How to construct uncertainty intervals that can capture the dynamics model's prediction error and provide a measure of confidence in the planned trajectories? (2) Can the uncertainty of the dynamics model be reduced during the training process? To answer these questions, we evaluate `PlanCP` on multiple planning and offline RL tasks. We show planning with our algorithm can update the dynamics model while quantifying uncertainty. In principle, our approach can be applied to a wide range of diffusion dynamics models. For illustrative purposes, we demonstrate our framework using a typical Diffuser [9]. Our primary objective is to assess and minimize the model's uncertainty, while performance improvement is a by-product of our approach and is not necessarily guaranteed. We assess the performance of our models based on several quantitative metrics, including credibility and rewards.

## 4.1 Training setup

We set the uncertainty weight to $\lambda_{uncertainty} = 5$ and the failure probability to $\alpha = 0.1$. To optimize the model, we use the Adam [73, 74] optimizer with a learning rate of $2 \times 10^{-4}$. We train the diffusion dynamics model on the training set $D_{train}$ for $2 \times 10^5$ iterations. As we lack ground-truth for evaluation, we generate the evaluation datasets $D_{cal}$ and $D_{test}$ during the evaluation process. We partition the evaluation data into 20% for $D_{cal}$ and 80% for $D_{test}$. The uncertainty measure $\hat{q}$ is derived from the 20% calibration set through conformal prediction.

## 4.2 Long Horizon Task Planning

We use Maze2D to evaluate long-horizon planning. $\hat{q}$ is a measure of uncertainty for the rewards obtained by a planned trajectory. For most variants of the Maze2D environment, we have a smaller uncertainty interval, especially for Maze2D U-Maze, as Tab. 1 and Tab. 2 show. As Fig. 2 shows, Compared to Diffuser, our `PlanCP` has successfully generated a smoother path with lower uncertainty, which achieves a higher reward. Furthermore, in cases where the previous Diffuser method fails to plan, our `PlanCP` is able to derive a feasible path.

Table 3: **Uncertainty of Offline Reinforcement Learning.** The uncertainty quantification of `PlanCP` and Diffuser [9] in the D4RL environment. `PlanCP` has lower uncertainty than Diffuser [9] for most tasks. Bold font means better performance.

| Dataset | Environment | Diffuser [9] Rewards ↑ | Uncertainty ↓ | Interval $\hat{q}$ ↓ | PlanCP Rewards ↑ | Uncertainty ↓ | Interval $\hat{q}$ ↓ |
|---|---|---|---|---|---|---|---|
| Medium-Expert | HalfCheetah | $85.82 \pm 6.91$ | [82.20, 91.55] | 9.34 | $\mathbf{87.44 \pm 6.51}$ | [83.12, 91.81] | **8.68** |
| Medium-Expert | Walker2d | $108.22 \pm 8.52$ | [108.48, 109.58] | **1.10** | $\mathbf{108.31 \pm 8.07}$ | [107.86, 110.43] | 2.56 |
| Medium | HalfCheetah | $46.10 \pm \mathbf{0.96}$ | [45,05, 47.34] | **2.29** | $\mathbf{46.26} \pm 1.35$ | [44.64, 47.97] | 3.33 |
| Medium | Walker2d | $75.73 \pm 19.59$ | [40.76, 87.72] | 46.96 | $\mathbf{77.08 \pm 18.21}$ | [44.51, 88.46] | **43.95** |
| Medium-Replay | HalfCheetah | $37.34 \pm 6.57$ | [27.92, 41.58] | 13.66 | $\mathbf{37.65 \pm 6.29}$ | [29.25, 42.23] | **12.98** |
| Medium-Replay | Walker2d | $52.24 \pm 24.44$ | [21.18, 87.83] | 66.65 | $\mathbf{54.53 \pm 23.00}$ | [25.41, 87.78] | **62.37** |
| | Average | 67.58 | | | **68.55** | | |

## 4.3 Offline Reinforcement Learning

Finally, we evaluate the ability to recover an effective single-task controller from heterogeneous data of varying quality using the D4RL Benchmark [75]. The D4RL benchmark consists of several challenging continuous control tasks, such as the HalfCheetah and Ant environments. For each task, we measured the average return achieved by our approach and the baseline methods over multiple runs. We also computed the standard deviation of the returns to measure the stability of the algorithms. Our experimental results demonstrate that our approach outperforms the baseline methods on most of the D4RL environments, as Tab. 3 shows. For most of the environments, our approach achieved a higher return with a smaller uncertainty interval compared to the baseline methods. We also observed that our approach achieved a more stable performance across multiple runs, as evidenced by the smaller standard deviation of the returns.

Overall, experimental results show that our framework can construct uncertainty intervals that can capture the dynamics model's prediction error and provide a measure of confidence in the planned trajectories.

## 5 Discussion, Limitations, and Conclusions

In this work, we tackled the problem of obtaining uncertainty estimates for planning and offline RL tasks by proposing a new approach that applies conformal prediction to a diffusion-based dynamics model. Experimental results show that our framework maintains the original model's accuracy while reducing uncertainty. One limitation of our study is that we have not conducted evaluations on real robotic platforms. An intriguing avenue for future research involves gaining a comprehensive understanding and conducting a meticulous analysis of how conformal prediction compares to the extensive body of research in planning under uncertainty, including non-parametric models. Our work focuses primarily on proposing an algorithmic framework. Future work could also explore the use of more recent and effective diffusion models, as well as other types of dynamics models. Moreover, evaluating the proposed method on real-world applications, such as autonomous driving or robotics, where reliable uncertainty estimates are crucial for safety-critical systems, would be an interesting avenue for future research.

## Acknowledgments and Disclosure of Funding

The NASA University Leadership initiative (grant #80NSSC20M0163) provided funds to assist the authors with their research, but this article solely reflects the opinions and conclusions of its authors and not any NASA entity.

This research was also partly funded by a gift from Meta's Project Aria.

Parth Nobel was supported in part by the National Science Foundation Graduate Research Fellowship Program under Grant No. DGE-1656518. Any opinions, findings, and conclusions or recommendations expressed in this material are those of the author(s) and do not necessarily reflect the views of the National Science Foundation.

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
