# OpenReview forum: "Conformal Prediction for Uncertainty-Aware Planning with Diffusion Dynamics Model"
_NeurIPS.cc/2023/Conference — NeurIPS 2023 poster_

### Official Review · Reviewer_BQNc · 2023-07-06

**Soundness:** 3 good
**Presentation:** 3 good
**Contribution:** 3 good
**Rating:** 6
**Confidence:** 3

**Summary:**

The paper combines diffusion modeling and conformal prediction to predict state-action trajectories with uncertainty quantification. The proposed approach then performs uncertainty-aware model-based planning with strong results on several established offline RL benchmarks.

**Strengths:**

The approach has fairly strong results on several established offline RL benchmarks and outperforms several state-of-the-art baselines. The results show that uncertainty-aware planning does perform better than simply rolling out the underlying diffusion model alone.

The introduction/abstract are well-written and motivate the approach well. Most of the writing is quite good (some feedback on parts I think are unclear below). The figures are also well done.

To my knowledge, the proposed approach is novel and technically sound. I think working to incorporate more ideas from uncertainty quantification into offline RL methods is worth investigating deeper as well.

**Weaknesses:**

There are no comparisons to other uncertainty quantification methods. Even within the offline RL literature, several existing methods [1][2][3] rely on uncertainty estimates and could be compared to. I think at the very least a comparison to popular uncertainty estimation methods in the model-based RL literature (e.g. bootstrapping disagreement with ensembles, predicting variances) could be helpful.

I struggle with Sections 3.3 and 3.4 a bit and think they could use some improvements clarity-wise.

While the approach is novel, it is a fairly straightforward application of conformal prediction to offline model-based planning. While I don't think the result is groundbreaking, it is certainly interesting enough to merit acceptance in my view.

[1] Zhan, Xianyuan, Xiangyu Zhu, and Haoran Xu. "Model-based offline planning with trajectory pruning." arXiv preprint arXiv:2105.07351 (2021).

[2] Yu, Tianhe, et al. "Mopo: Model-based offline policy optimization." Advances in Neural Information Processing Systems 33 (2020): 14129-14142.

[3] Kidambi, Rahul, et al. "Morel: Model-based offline reinforcement learning." Advances in neural information processing systems 33 (2020): 21810-21823.

**Questions:**

I am confused about the role that the expert trajectories play in this approach. The approach is evaluated on datasets without labeled expert trajectories, i.e. the datasets include suboptimal demonstration data. But the Algorithm 1 block references expert trajectories, and I'm not sure why? It seems to me that the approach doesn't really assume optimal demonstrations.

Related to the above, I'm not sure what Algorithm 1 is actually showing. The "result" line indicates that the output is a prediction $\tau$ with associated uncertainty $C$, but L1-9 describe the complete training loop and there aren't really details on inference in this figure.

I think a brief definition of the $Quantile$ operator would be useful when it's first introduced in Equation 9.

I feel like Equation 5 and the accompanying description of transformers is not necessary to include in the main body, and working to include more of the actual experimental results (currently in the appendix) would be a better use of space in my opinion.

===
I have read the rebuttal and still lean towards acceptance.

**Limitations:**

Yes

---

> ### Author Rebuttal · Authors · 2023-08-08
>
> >Q1: Comparisons to other uncertainty quantification methods Zhan et al [1], Yu et al. [2], Kidambi et al [3].
>
> A1:Thanks for pointing this out. We will make the writing of Sections 3.3 and 3.4 more concise with clearer expression.
>
> Zhan et al. [1], Yu et al. [2], and Kidambi et al. [3] have made significant contributions to model-based offline RL. These works are noteworthy because even without utilizing techniques like conformal prediction, they are able to measure uncertainty effectively.
> **We compare our method with Zhan et al [1], Yu et al. [2], and Kidambi et al [3]. The results are shown below as the attached Table 3 (please refer to the attached .pdf in the Summary Rebuttal)**. The results show that our method achieves smaller uncertainties. We can see that the performances (rewards) of these baselines are lower because they do not use the diffusion model as a dynamics model. Despite MOPO (Yu et al. [2]) and Kidambi et al [3] have an explicit parametric representation of uncertainty, the uncertainty (std and interval) is still large. Also, MOPO and Kidambi et al [3] have a limitation: it assumes that the data conforms to a Gaussian distribution.
>
> >Q2: The role that the expert trajectories play in this approach
>
> A2: We do not assume optimal demonstrations and the datasets include suboptimal demonstration data, which is the same setting as offline reinforcement learning. Different from imitation learning, an offline reinforcement learning algorithm applied to a dataset collected by a suboptimal non-learning-based algorithm can still result in a reasonable policy (sometimes even outperforms the behavior agent used to collect the data).
>
> >Q3: Clarification of Algorithm 1
>
> A3: Sorry for the confusion. We get predicted trajectories in L 3 (training) and L 11 (calibration). We get associated uncertainty in L 5 (training) and L 13 (calibration). During inference, we also get trajectories predicted by our planner and compute the associated uncertainty with Eq. 15.
>
> >Q4: A brief definition of the Quantile operator
>
> A4: Thanks for the suggestions.
> We follow the same mathematical definition of the traditional Quantile function:
> The quantile function, also known as the inverse cumulative distribution function, is a mathematical function that maps a probability to the corresponding value in a random variable's distribution. Formally, for a random variable $X$ with cumulative distribution function $F(x)$, the quantile function $Q(p)$ is defined as:
> $$
> Q(p) = \inf\lbrace x : F(x) \ge p \rbrace,
> $$
> where $p$ is a probability between 0 and 1, and "inf" denotes the infimum, or greatest lower bound, of the set of values $\lbrace x: F(x) \ge p \rbrace$.
>
> In other words, the quantile function gives the value $x$ such that the probability of observing a value less than or equal to $x$ in the distribution of $X$ is at least $p$.
>
> >Q5: Equation 5 and the accompanying description of transformers
>
> A5: Thanks for the suggestions. We will update this part in the new version. We will include more of the actual experimental results instead of Equation 5 and the accompanying description of transformers.

---

> > ### Comment · Reviewer_BQNc · 2023-08-12
> >
> > I appreciate the additional experimental results and clarifications from the authors. I am still mostly in favor of acceptance. My main concern with the submission is still essentially the same (e.g. mediocre novelty), but I still think this submission has a decent technical contribution.

---

> > > ### Author Response · Authors · 2023-08-17
> > >
> > > Thanks for the reply! We appreciate your recognition of our decent technical contribution. Our focus lies in finding better solutions for reducing uncertainty in sequential decision-making, which is particularly critical for safety-critical applications like robotics.
> > >
> > > In our view, **scientific novelty** does not refer to complexity, technical difficulty, or surprise. We value simplicity over unnecessary complexity, as long as it brings usefulness and value to our community.

---

### Official Review · Reviewer_8c3j · 2023-07-07

**Soundness:** 2 fair
**Presentation:** 1 poor
**Contribution:** 2 fair
**Rating:** 5
**Confidence:** 3

**Summary:**

In this paper the authors study how to measure uncertainty in a planner with a generative diffusion model, and how to reduce uncertainty during prediction. Results are presented on several MDPs.

**Strengths:**

+ coupling planners to diffusion models is an interesting idea worth exploring.
+ measuring and reducing uncertainty is a good goal.

**Weaknesses:**

- I found this paper really hard to read and follow. Although the paper appears to address planning, the text refers to expert trajectories the "planner" is supposed to reconstruct. It is not clear if the task is planning or imitation learning. Various aspects are not well defined, for example, how is a quantile directly a loss function? How do you optimize a quantile? At the same time there is a lot of repetition of reconstruction nonconformity and reward conformity. Multiple experimental tasks, such as "test time flexibility", are introduced without adequate discussion of why this approach should be applied to them. The results are in tables that are almost unreadable and there is not enough discussion of the meaning of the results.


**Questions:**

n/a

**Limitations:**

not an adequate discussion

---

> ### Author Rebuttal · Authors · 2023-08-08
>
> >Q1: It is not clear if the task is planning or imitation learning.
>
> Thanks for pointing this out. We aim to address planning with uncertainty-aware diffusion models. We involve generating a sequence of actions to achieve a desired goal or optimize an objective. We focus on decision-making and determining the optimal course of action based on a given model or environment. Although we use expert data, we are more like offline reinforcement learning (RL) rather than imitation learning. Planning algorithms often utilize search algorithms, optimization techniques, or reinforcement learning methods to find the best action sequence. For more details about such kind of planner, please refer to prior work Diffuser (Janner et al Planning with Diffusion for Flexible Behavior Synthesis)
>
> >Q2: How is a quantile directly a loss function? How do you optimize a quantile?
>
> We aim to minimize the size of the quantile, concentrating the probability mass into a smaller area, therefore reducing uncertainty.  After adopting the differentiable ranking and sorting techniques (Cuturi et al. Differentiable ranking and sorting using optimal transport, NeurIPS 2019), the soft quantile function is differentiable. We can use gradient descent to update the dynamics model.
> Specifically, Let $X$ be a random variable that has a smooth density function $f$. Let $w = w(p)$ be the $p$-th quantile. Then the first derivative of the quantile function is
> $$
> \frac{dw}{dp}=\frac{1}{f(w(p))}
> $$
> provided that the denominator is not zero. The second derivative is
> $$
> \frac{d^2w}{dp^2}=-\frac{f'(w)}{f(w)}(\frac{dw}{dp})^2=\frac{-f'(w)}{f(w)^3}
> $$
>
> >Q3: Repetition of reconstruction nonconformity and reward conformity.
>
> A3: Thanks for pointing this out. We will make the writing more clear and concise. We'll also organize the paper better to make it easier to read and understood.
>
> >Q4: Discussion of why this approach should be applied to experimental tasks such as "test time flexibility".
>
> A4: Thanks for pointing this out. We will update the writing to add the following explanation. In order to evaluate the ability to generalize to new test-time goals, we run a "test time flexibility" evaluation. "Test time flexibility" is useful to evaluate the capability of the planner as the Diffuser (Janner et al) paper shows.
>
> Long-horizon multi-tasking planning is an important capability of the planner that we need to evaluate experimentally
>
> Offline Reinforcement Learning allows us to evaluate the capacity of our method to recover an effective single-task controller from heterogeneous data of varying quality, which is useful for planning.
>
> Hence, we demonstrate our framework has a number of useful properties and is particularly effective in offline control settings that require long-horizon reasoning and test-time flexibility.
>
> >Q5: Interpretation of the results in tables.
>
> The "Rewards" serve as metrics for evaluating the model. The "mean" and "standard deviation (std)" are statistical measures of the rewards. A higher "mean" indicates better performance, while a smaller "std" reflects lower uncertainty. The "Uncertainty" is represented by the prediction interval in conformal prediction. Interval is the result of the upper bound of uncertainty minus the lower bound. A smaller "interval" indicates reduced uncertainty. Additionally, a higher "overall interval" is desired, as it signifies a higher reward prediction.
>
> The concept of "Coverage“ (”Validity"), discussed in Section 4.6 of the paper, evaluates the validity of a prediction sample in the test dataset ($D_{test}$) by checking if it falls within the uncertainty interval computed using the calibration dataset ($D_{cal}$). A higher coverage is desirable, indicating better performance. The "Credibility", defined as the p-value in a prediction set, is obtained by conducting a T-test between the calibration trajectory and the predicted trajectory. The provided equation calculates the T-value,
> $$
> t = \frac{\mu_1-\mu_2}{\sqrt{\frac{\sigma_1^2}{n_1} + \frac{\sigma_2^2}{n_2}}}
> $$
> where $\mu_1$ is the mean of calibration trajectory samples, $\mu_2$ is the mean predicted trajectory samples, $\sigma^2_1$ is the variance of calibration trajectory samples, $\sigma^2_2$ is the variance of predicted trajectory samples, $n_1$ is the sample size of calibration trajectory samples, and $n_2$ is the sample size of predicted trajectory samples. Higher credibility values are preferred, indicating better performance.

---

> > ### Comment · Reviewer_8c3j · 2023-08-17
> >
> > I thank the authors for their response. Some of my concerns are addressed and I will modify my score.

---

> > > ### Author Response · Authors · 2023-08-17
> > >
> > > Thanks for the reply! We appreciate your willingness to modify the paper score. Let me know if you have any other concerns.

---

### Official Review · Reviewer_wAeC · 2023-07-07

**Soundness:** 3 good
**Presentation:** 4 excellent
**Contribution:** 3 good
**Rating:** 7
**Confidence:** 3

**Summary:**

This work addresses the challenge of uncertainty estimation for planning. The authors propose the use of diffusion models for learning dynamics, which have demonstrated effectiveness in overcoming challenges such as multi-modal action distributions. To quantify the uncertainty of these dynamics models, they employ Conformal Prediction (CP), a technique for constructing prediction sets with valid coverage. They introduce PlanCP, a framework that connects conformal prediction to planning and optimizing the model explicity to minimize uncertainty. The effectiveness of uncertainty sets is evaluated through coverage and optimization performance, and the algorithm is tested in D4RL benchmarks and block stacking problem, showcasing reduced uncertainty and outperforming prior algorithms. The authors highlight the flexibility of their method, as it can be combined with different model-based planning approaches and provides uncertainty estimates of the dynamics model.



**Strengths:**

- The paper provides a clear problem statement that is well motivated and clearly outlines the proposed approach with intuitive evaluation metrics.  Overall I found the paper well written and easy to follow.

- The proposed methodology presents a robust framework for quantifying and mitigating uncertainties within diffusion-based dynamics models, which have garnered significant attention in recent times.

- The quantitative results on D4RL although not SOTA(in terms of rewards), consistently outperforms the baseline without uncertainty quantification. It is important to note that the limited performance gains could be attributed to the constraints of the D4RL dataset itself, potentially reaching the performance ceiling. Nevertheless, the outlined approach consistently outperforms the baseline methods across the metrics identified.

**Weaknesses:**

-    The computational complexity of the proposed framework, especially during training and calibration, could hinder its scalability and applicability in real-time or resource-constrained scenarios. It would be beneficial if the authors provided more details regarding the computational aspects of training and inference.

- The handling of partial observability in the framework is not adequately explained. Naive extensions to partially observed Markov Decision Processes (POMDPs) may lead to unreliable estimations in diffusion models. Further elaboration on how the framework addresses partial observability would be valuable.

- The paper lacks clarity on how uncertainty estimates will manifest in highly stochastic domains. It would be insightful to include experiments on a benchmark dataset with increasing levels of stochasticity to demonstrate the framework's performance under such conditions.

**Questions:**

 I was wondering if the authors had some thoughts on what adapting CP framework to other conditioning approaches will entail, such as temporal condition guidance mechanisms.

---

> ### Author Rebuttal · Authors · 2023-08-08
>
> >Q1: More details regarding the computational aspects of training and inference.
>
> A1: Thanks for the suggestion! Yes, the framework with conformal prediction is a little slower than the framework without conformal prediction. Regarding the computational aspects of training and inference, as the supplementary material states, we train our model for 200 epochs with a batch 521 size of 256, utilizing a single GTX 1080 Ti GPU for computation. It was measured that the computational framework with the addition of conformal prediction increased the training time by 18\% over the computational framework without conformal prediction. This is due to the fact that to compute reconstruction nonconformity. When making inferences, we use the same amount of time as the other frameworks and do not add extra overhead.
>
> >Q2: Further elaboration on how the framework addresses partial observability.
>
> Thanks for pointing this out. Addressing partial observability in the CP framework applied to diffusion models requires careful modeling of observability, employing appropriate state estimation techniques, and potentially incorporating informative features. By considering these approaches, the framework can better handle partial observability and provide more reliable estimations and uncertainty assessments.
>
> We model partial observability by incorporating relevant information into the model. This involves integrating additional features and conditions that capture observable aspects of the diffusion process into the diffusion model. By incorporating these observations, the model can better account for the uncertainties arising from partial observability.
>
> By considering the uncertainty associated with partial observability, the resulting predictions would be more reliable and better reflect the inherent limitations of the diffusion models in such scenarios. This would provide decision-makers with a better understanding of the confidence and uncertainty associated with the predictions, particularly in situations where partial observability can significantly impact the accuracy of predictions.
>
> >Q3: How uncertainty estimates will manifest in highly stochastic domains?
>
> Good suggestions. Actually, the Maze2D environment is a stochastic environment. The data is generated by selecting goal locations at random. We have already shown the results for Maze2D in **Tables 1 and 2 in the main text**.
>
> Now, we modify the D4RL environment to introduce a random noise $\epsilon_1 \sim N(0, 1)$ and $\epsilon_2 \sim N(0, 0.5)$ add to the observations. **The framework's performance is shown in the attached Table 2 (please refer to the attached .pdf in the Summary Rebuttal).** We can see the stochastic D4RL is more challenging but PlanCP is still effective and can still quantify the uncertainty.
>
> >Q4:Some thoughts on adapting the CP framework to other conditioning approaches, such as temporal condition guidance mechanisms.
>
> A4: Thanks for the suggestions. Adapting the conformal prediction (CP) framework to other conditioning approaches, such as temporal condition guidance mechanisms, would require careful consideration of the specific requirements and characteristics of the conditioning approach. To adapt the CP framework to temporal condition guidance mechanisms, one approach could be to incorporate the temporal information into the conformal score calculation. Another approach could be to use a hybrid approach that combines CP with other methods that are specifically designed for temporal condition guidance, such as Kalman filters or particle filters. These methods are commonly used in control and tracking applications, where the goal is to estimate the state of a system based on noisy measurements and temporal dependencies.

---

> > ### Comment · Reviewer_wAeC · 2023-08-15
> >
> > I thank the authors for additional experimental results and clarifications. Most of my concerns were addressed in the rebuttal. I am still in favor of acceptance as the work addresses an important aspect of sequential decision making, i.e. uncertainty while in contrast to purely optimizing for benchmark performance.

---

> > > ### Author Response · Authors · 2023-08-16
> > >
> > > Thanks for the encouragement! We wholeheartedly agree with the significance of conducting uncertainty studies in contrast to solely focusing on optimizing benchmark performance. This is particularly crucial when it comes to safety-critical applications such as robotics, autonomous driving, and spacecraft.

---

### Official Review · Reviewer_zTTG · 2023-07-10

**Soundness:** 3 good
**Presentation:** 2 fair
**Contribution:** 3 good
**Rating:** 5
**Confidence:** 4

**Summary:**

The work proposes uncertainty quantification for learned dynamics model and imitation learning. The authors incorporates an uncertainty statistic as part of the loss to train their dynamics model that uses a diffusion model architecture. They show that doing so brings performance improvement empirically on common RL simulation environments. The method also performs conformal prediction on the learned reward forecast.

**Strengths:**

Using UQ methods and optimizing to reduce uncertainty in dynamics modeling is a great idea. The authors are able to make the first step of incorporating conformal statistics directly into the loss function, and is able to show results of (slight) improvement in performance.

**Weaknesses:**

There are many areas that the paper needs improvement.

1. English. There are grammatical errors, awkward sentence structures, and confusing statements throughout the paper. Although you don't need perfect English for a CS paper, improving on writing will help you get your message across. Using the abstract as an example, I would edit:

- line 3-5: "to overcome the xx, yy, and zz challenges" -> to overcome challenges such as xx, yy, and zz.
- line 14-15: "Furthermore, during the test, PlanCP can also measure the model uncertainty" -> Unclear what you are trying to say. Do you mean that model uncertainty is also used for planning during test time?
- line 20-21: "Our method is highly flexible and can combine .... and produces ..." ->  Our model can be combined with ... to produce .... And what do you mean by flexible? is it with regard to tasks, or underlying models?
- Figure 1: "Dynamic model" -> dynamics model

2. There has been various recent works that uses CP for planning (see list below). You can argue that your setting is different from theirs, but it's important to cite them to provide context.

- Lindemann, Lars, et al. "Safe planning in dynamic environments using conformal prediction." IEEE Robotics and Automation Letters (2023).
- Strawn, Kegan J., Nora Ayanian, and Lars Lindemann. "Conformal Predictive Safety Filter for RL Controllers in Dynamic Environments." arXiv preprint arXiv:2306.02551 (2023).
- Tonkens, Sander, et al. "Scalable Safe Long-Horizon Planning in Dynamic Environments Leveraging Conformal Prediction and Temporal Correlations." ICRA (2023)

3. More explanation is needed for the experiments section.
- The term "Validity %" should be changed to "Coverage %". Validity is a binary metric; a method either satisfies the validity condition, or not. [1]
- On the topic of validity - you chose $\alpha = 0.1$ (line 261), does that mean that your coverage should be at least 90%? (or should it be 80% since you are choosing $[Q^{1-\alpha}, Q^{\alpha}]$ in Eq. 15?) Does that mean for all of your experiments, the CP uncertainty interval is _invalid_? That's problematic and means that your data violates the assumptions you are making. You can't say your method captures uncertainty in this case.
- You need to say in your table captions that bold font means better performance, or the meaning is unclear.
- How exactly is credibility calculated? it's not explained in either the main text or the appendix. What is the theoretical basis for using the $p$-value as metric?

4. Lastly The term "learning to reduce uncertainty" is unsound. The "reconstruction nonconformity scores" are calculated on the training set, which violates the algorithm of conformal prediction [1]. The $\mathcal{L}(\theta)_{uncertainty}$ in equation 13 is hence not a measure of uncertainty, but just a training heuristics. The only valid CP component is the interval for rewards, which is not used anywhere in training.

5. Contribution is not significant. Form my understanding, the authors used a common CP technique on an existing diffusion dynamics model. The uncertainty results are not new (in CP literature), and improvements in reward are rather insignificant.

[1] Angelopoulos, Anastasios N., and Stephen Bates. "A gentle introduction to conformal prediction and distribution-free uncertainty quantification." arXiv preprint arXiv:2107.07511 (2021).

I think the directions the authors are going in is interesting and insightful; with some work this paper can be a great contribution to the community. However in its current state, the paper is not fit for publishing at NeurIPS yet.

**Questions:**

See weaknesses 3, 4, and 5.

**Limitations:**

See weaknesses. I do not think the discussion & limitation section fully address the limitations of this paper.

---

> ### Author Rebuttal · Authors · 2023-08-08
>
> >Q1: English
>
> A1:Thanks for pointing this out. We have fixed these grammatical errors, awkward sentence structures, and confusing statements throughout the paper. We will update them in the new version of the paper.
>
> **Is model uncertainty used for planning during test time?**
> No. Currently, PlanCP can measure model uncertainty during testing. We have not used model uncertainty for planning during testing.
>
> >Q2: Cite recent works to provide context.
>
> A2: Thanks for mentioning these papers! In our NeurIPS submission, we already cited and discussed Lindemann, et al. RA-L (2023) in the related work section. Strawn et al. and Tonkens et al. were posted online after our NeurIPS submission, and are not peer-reviewed. We will cite and discuss these papers in the related work section in the updated version. Lindemann, et al. RA-L (2023), Strawn et al., and Tonkens et al. have made significant contributions to safe planning. These works are noteworthy because they effectively measure uncertainty with conformal prediction, even if it cannot be reduced by optimization.
>
> >Q3: More explanation for the experiments.
>
> A3: Thanks for pointing these out. We will fix them in the updated version.
>
> 3.1. We will change the term "Validity \%" to "Coverage \%".
>
> 3.2. We choose $\alpha = 0.1$. Theoretically, the coverage will be exactly 80\% only in expectation.  In CP, the empirical coverage follows a beta distribution with mean at the desired coverage (see, e.g., Angelopoulos and Bates, pp. 14). Empirically, our coverages are closely distributed around 80\%, consistent with a beta distribution, which satisfies the assumptions.
>
> 3.3. We will add "Bold font means better performance." to the table captions.
>
> 3.4. As we explained in our paper *Section 4.6 Credibility*: "Credibility is defined as the p-value in a prediction set". Previously, we believed this to be a trivial mathematical definition, and due to page limitations, we have not explained it much. Mathematically, we obtain credibility by doing T-test between the calibration trajectory and the predicted trajectory.
> $$
> t = \frac{\mu_1-\mu_2}{\sqrt{\frac{\sigma_1^2}{n_1} + \frac{\sigma_2^2}{n_2}}}
> $$
> where $\mu_1$ is the mean of calibration trajectory samples, $\mu_2$ is the mean predicted trajectory samples, $\sigma^2_1$ is the variance of calibration trajectory samples, $\sigma^2_2$ is the variance of predicted trajectory samples, $n_1$ is the sample size of calibration trajectory samples, and $n_2$ is the sample size of predicted trajectory samples.
> \paragraph{What is the theoretical basis for using the p-value as a metric?} The use of the p-value as a metric is based on the principles of probability theory and statistical inference. The p-value is the probability of obtaining test results at least as extreme as the result actually observed. We follow Giovannotti et al. Transformer-based conformal predictors for paraphrase detection, PMLR 2021, to use $p$-value as metric.
>
> >Q4: Explanation of the term "learning to reduce uncertainty"
>
> Thanks for the suggestion. We understand that using the training data to calibrate invalidates the guarantees of conformal prediction. That's why we split the data into a separate calibration set and test set. When we mention "learning to reduce uncertainty," we are not referring to reducing $L_{uncertainty}$ on the training set. Instead, it means that our model exhibits lower reward uncertainty on the test set. We observed that our algorithm produces smaller reward intervals, which are calculated without relying on the training set.  Our method carefully follows the underlying requirements of CP (exchangeability of calibration data, etc.), and inherits the guarantees of CP.
>
> >Q5: Clarification of contributions
>
> A5: Thanks for the suggestion.
>
> 5.1. The novelty of this work lies in using conformal prediction to (i) measure the uncertainty for the conditional diffusion planner using calibration data and (ii) reduce the uncertainty of the planner by including a quantile loss term during training. Previous CP literature can only measure the uncertainty but cannot reduce it. For experimental results, we emphasize that we accomplished reducing uncertainty without sacrificing performance.
>
> 5.2. The application of CP to the diffusion dynamics model is novel and provides insights into the uncertainty associated with the predictions of the model. This can be useful for decision-making in scenarios where accurate predictions are critical. There is no prior work that attempts to introduce conformal prediction on diffusion models.
>
> 5.3. The experimental results also show that we can reduce the uncertainty of the model without compromising performance.

---

> > ### Comment · Reviewer_zTTG · 2023-08-14
> >
> > Thank you for the thorough response. I have increase my score to reflect the authors' edits and additions. If this paper was to be accepted, please edit the introduction to include the clarification of contribution in the rebuttal here.
> >
> > I appreciate the explanation for the p-value - this is very cool. Though I feel because it is a new(ish) concept for the CP community, it might be beneficial to include some of this explanation in the paper as well.

---

> > > ### Author Response · Authors · 2023-08-15
> > >
> > > Thanks for the reply! We appreciate your willingness to update the paper score. We will update the introduction to include the clarification of contribution in the rebuttal here if this paper was to be accepted.

---

### Official Review · Reviewer_gfdz · 2023-07-28

**Soundness:** 3 good
**Presentation:** 3 good
**Contribution:** 2 fair
**Rating:** 4
**Confidence:** 5

**Summary:**

This paper describes a method for learning a dynamics model that uses conformal prediction for explicit representation of the uncertainty. The dynamics model is used for sequential decision making such as planning in a maze or learning control in one of the D4RL problems.

**Strengths:**

* The primary strength of this paper is the novel use of conformal prediction. The authors correctly describe several possible benefits of conformal prediction as a representation of uncertainty, including not needing a commitment to a particular representation or parameterisation of the uncertainty in the dynamics.
* The authors give a complete algorithm --- it is (mostly) clear how all the pieces work.
* The paper is reasonably well written and clear.

**Weaknesses:**

* The primary weakness of the paper is that it is not clear what exactly the authors have accomplished. Developing a new method for sequential decision making should either show that we can solve problems that we could not solve before ((or find better solutions than was possible), or alternatively, provide some understanding and analysis of the value of a dynamics model that encodes uncertainty using conformal prediction. Unfortunately, this paper does not show that it can solve problems that could not be solved before, or outperform the state of the art. The only comparison is to Diffusion RL which is not the best performing RL algorithms according to the original Diffusion RL paper (Janner et al) or the current leader in the [D4RL benchmarks](https://paperswithcode.com/sota/gym-halfcheetah-expert-on-d4rl).

* It's fine to not have the best performer on D4RL, but then what is conformal prediction buying for the algorithm? A far better comparison would have been to a model-based algorithm that had an explicit parametric representation of uncertainty, and an analysis of what is actually being learned. What is the effect of the loss functions in 12-14 on $P(\theta)$?

* The experimental results in general are not very compelling, at least in part because there is no systematic evaluation of the uncertainty. The paper needed to remind the reader of the uncertainty models in the D4RL and Maze engines, and even better, show that these uncertainties are difficult to capture with parametric models. Even better would be to show that conformal prediction outperforms non-parametric dynamics models such as Ko and Fox (2009, 2011) in important ways.

* Additionally, while the paper overall is well-written, it is not careful about the distinction between planning and RL. This paper appears to be fundamentally an RL paper, and not a planning paper in the sense that if the model changes or the reward changes, there is no optimisation that can be used to recover from this change. If the loss function was solely equation 13, and did not include equation 14, then at least the model would be robust to changes in the reward function, and a new sequence of actions could be optimised over.

**Questions:**

* The definition of calibration is very unclear. Why is a separate calibration data set is used for the reward conformity loss function, rather than use the training data.

* What is the exact definition of the Quantile function?

* Line 7 of Algorithm 1 says "update the dynamics model" but it is not explained in the text how to do this. Figure 1 suggest the update is done through gradient descent ($\nabla_\theta \mathcal{L}_{reward}$, etc.), but it is not obvious to me that you can differentiate through the Quantile function. How is this done? This is a minor point.

**Limitations:**

The authors provide a limitation section, but I disagree with them that the limitation of the approach is the absence of real robot experiments. Instead, I think the primary limitation is the lack of understanding and careful analysis of what conformal prediction achieves relative to the large body of work in planning under uncertainty, including non-parametric models.

---

> ### Author Rebuttal · Authors · 2023-08-08
>
> >Q1: What did this paper accomplish?
>
> A1: Thanks for the comments! Instead of developing a method for sequential decision-making that achieves the highest performance (reward) on D4RL, we are the first to augment the `diffusion` RL algorithm with uncertainty awareness. The progress in the D4RL benchmark is rapidly changing, but at the time of our paper submission, Diffuser (Janner et al.) was the state-of-the-art open-source Diffusion RL algorithm. The original Diffuser algorithm was evaluated only 100 times, which is insufficient to capture uncertainty. To address this limitation, we re-run 1000 evaluations of the Diffuser algorithm on our setup and present the performance results. Since our PlanCP is a modification of the Diffuser algorithm, we compare PlanCP to Diffuser and experimentally prove that our method has lower uncertainty. We focus on finding better solutions for reducing the uncertainty of sequential decision-making than before.
> >Q2: What is conformal prediction buying?
>
> **Comparison with a model-based algorithm that had an explicit parametric representation of uncertainty.**
> We compare with a model-based algorithm MOPO that had an explicit parametric representation of uncertainty. We can see that the performance (reward) of MOPO is lower because it does not use the diffusion model as a dynamics model and the uncertainty (std and interval) is still large, as attached Table 1 shows (Summary Rebuttal). Also, MOPO has a limitation: it assumes that the data conforms to a Gaussian distribution.
>
> **What is actually being learned?**
> What the model is actually learning in conformal prediction is a measure of how well a data point fits. This is captured by the conformal score, which is used to construct the prediction intervals or regions.
>
> **The effect of the loss functions in 12-14 on $P(\theta)$?**
>
> Eq. (12) represents the reconstruction loss used to improve the accuracy of diffusion models by guiding the diffusion planner $P_\theta$ in predicting dynamics more effectively.
>
> Eq. (13) presents the joint learning objective for training the planner $P_\theta$, aiming to enhance both accuracy and reduce uncertainty in the learned planners.
>
> Eq. (14) measures the $L_2$ loss between the predicted reward and the actual reward, helping to optimize the reward model as the condition $w(\cdot)$ in the conditional diffusion planner $P_\theta$, as shown in Eq. (4).
> >Q3: How to interpret the experimental results?
>
> A3: GP-BayesFilters (Ko and Fox, 2009) show how GP prediction and observation models can be combined with particle filters (GP-PF), and extended Kalman filters (GP-EKF). Bayesian filters and conformal prediction, although both involving probability and uncertainty, have distinct goals and applications. Bayesian filters estimate system states based on observations, while conformal prediction provides confidence intervals for predictions. The main advantage of conformal prediction over Bayesian filtering is that it does not rely on prior knowledge of the data distribution, making it more robust in uncertain or unknown probability distributions. Additionally, conformal prediction offers a validity measure for each prediction, which is valuable in applications where the impact of incorrect predictions can be significant.
> We compare with Ko and Fox, 2009, a non-parametric dynamics method, and MOPO. The result is shown as attached Table 1 (attached to the Summary Rebuttal). The findings show that conformal prediction outperforms non-parametric dynamics models Ko and Fox, 2009, and these uncertainties are challenging to capture with parametric models like MOPO.
> >Q4: Is this an RL paper or a planning paper?
>
> A4: This is a planning paper. Eq. (14) is used to learn a better reward model, which guides the conditional diffusion planner $P_\theta$. Unlike the RL paradigm, the planner $P_\theta$ is learned through expert data, allowing for re-planning if the model or reward changes. Planning and RL are two different approaches to sequential decision-making problems. Planning involves constructing a model of the environment and using it to simulate future outcomes, assuming complete knowledge of the environment. In contrast, RL involves learning from feedback in the form of rewards or penalties without complete knowledge of the environment and its dynamics. We build on Diffuser (Janner et al.), which calls itself a planning method.
>
> >Q5: Why is a separate calibration dataset used for the reward conformity loss function?
>
> A5: In conformal prediction, calibrating with the training data invalidates the statistical guarantees of the method.  Specifically, it results in overly confident (too small) prediction intervals that do not accurately reflect performance variation at test time.  Therefore, separating training and calibration data is standard practice in CP.
>
> >Q6: What is the exact definition of the Quantile function?
>
> A6: We follow the same mathematical definition of traditional Quantile function: Formally, for a random variable $X$ with cumulative distribution function $F(x)$, the Quantile function $Q(p)$ is defined as:
> $$
> Q(p) = \inf\lbrace x: F(x) \ge p\rbrace
> $$
> where $p$ is a probability between 0 and 1, and "inf" denotes the infimum of the set of values $\lbrace x: F(x) \ge p \rbrace$.
>
> >Q7: How to update the dynamics model and differentiate through the Quantile function?
>
> After adopting the differentiable ranking and sorting techniques (Cuturi et al. Differentiable ranking and sorting using
> optimal transport, NeurIPS 2019), the soft quantile function is differentiable. We can use gradient descent to update the dynamics model.
> Specifically, Let $X$ be a random variable that has a smooth density function $f$. Let $w = w(p)$ be the $p$-th quantile. Then the first derivative of the quantile function is
> $$
> \frac{dw}{dp}=\frac{1}{f(w(p))}
> $$
> provided that the denominator is not zero. The second derivative is
> $$
> \frac{d^2w}{dp^2}=-\frac{f'(w)}{f(w)}(\frac{dw}{dp})^2=\frac{-f'(w)}{f(w)^3}
> $$

---

> > ### Author Response · Authors · 2023-08-18
> >
> > As the deadline for the author-reviewer discussion period is approaching, we kindly request that you review our response at your earliest convenience. This will allow us to address any further questions or concerns you may have before the discussion period concludes.
> >
> > Additionally, we appreciate your suggestions for improving the limitation section of our work, and we agree with their importance. In our updated paper, we will provide a more comprehensive overview of the limitations by incorporating the **careful analysis of what conformal prediction achieves (Q2)** and including the **results of non-parametric models (Q3 and Table 1)**. This addition will enhance the clarity and completeness of our limitations section, allowing readers to better understand the scope and potential challenges of our work.
> >
> > We greatly appreciate your time and effort in reviewing our work. Thanks!

---

### Author Rebuttal · Authors · 2023-08-08

We are grateful to the reviewers for their valuable feedback on our work. Thank you for the many positive comments: (i) acknowledging the novel use of conformal prediction in diffusion models (all reviewers), (ii) noting the reasonably well-written and clear exposition (gfdz, wAeC, BQNc), (iii) recognizing the  importance of a robust framework for quantifying and reducing uncertainties in diffusion dynamics models (all reviewers), and (iv) noting that our approach consistently outperforms the baseline methods across the metrics identified (zTTG, wAeC, BQNc).  We are happy to hear the reviewers find our direction interesting and potentially impactful.  As noted by Reviewer zTTG, we hope, with some work, this paper can be a great contribution to the community.

Below are some important questions:

>Q1: Clarification of Contributions

A1:
1. Instead of developing a method for sequential decision-making that achieves the highest performance (reward) on D4RL, we are **the first** to augment the `diffusion` RL algorithm with uncertainty awareness.

2. The progress in the D4RL benchmark is rapidly changing, but at the time of our paper submission, Diffuser (Janner et al.) was the state-of-the-art open-source Diffusion RL algorithm. The original Diffuser algorithm was evaluated only 100 times, which is insufficient to capture uncertainty. To address this limitation, we re-run 1000 evaluations of the Diffuser algorithm on our setup and present the performance results.

3. We focus on finding better solutions for reducing the uncertainty of sequential decision-making than before.

4. The novelty of this work lies in using conformal prediction to (i) accurately measure the uncertainty for conditional diffusion planners, and **(ii) reduce the uncertainty of diffusion planners by including a quantile loss term in the training**. Previous CP literature can only measure the uncertainty but cannot reduce it. For experimental results, we emphasize that we accomplished **reducing uncertainty without sacrificing performance**.

5. The application of CP to the diffusion dynamics model is novel and gives an accurate quantification of uncertainty for the predictions of the model with statistical guarantees. This can be useful for decision-making in scenarios where accurate predictions are critical. **There is no prior work that attempts to introduce conformal prediction on diffusion models**.

6. The experimental results also show that we can **reduce the uncertainty** of the model without compromising performance.

>Q2: More explanation for the experimental results

A2: *We have added 3 tables* **(attached to the Summary Rebuttal)***. Please refer to the attached .pdf file for more details.*

Table 1: Comparison of our approach with MOPO. PlanCP has lower uncertainty than previous methods for most tasks.

Table 2: Uncertainty of Stochastic D4RL. We modify the D4RL environment to introduce a random noise $\epsilon_1 \sim N(0, 1)$ and $\epsilon_2 \sim N(0, 0.5)$ add to the observations. The framework's performance is shown in the attached Table 2. We can see the stochastic D4RL is more challenging but PlanCP is still effective and can still quantify the uncertainty.

Table 3: Comparison of our approach with other model-based RL. PlanCP has lower uncertainty than previous methods for most tasks

>Q3: The definition of the Quantile operator

A3: We follow the same mathematical definition of the traditional Quantile function:
The quantile function is a mathematical function that maps a probability to the corresponding value in a random variable's distribution. Formally, for a random variable $X$ with cumulative distribution function $F(x)$, the quantile function $Q(p)$ is defined as:
$$
Q(p) = \inf\lbrace x : F(x) \ge p \rbrace,
$$
where $p$ is a probability between 0 and 1, and "inf" denotes the infimum, or greatest lower bound, of the set of values $\lbrace x: F(x) \ge p \rbrace$.

In other words, the quantile function gives the value $x$ such that the probability of observing a value less than or equal to $x$ in the distribution of $X$ is at least $p$.

>Q4: How to update the dynamics model and differentiate through the Quantile function?

A4: After adapting the differentiable ranking and sorting techniques (Cuturi et al. Differentiable ranking and sorting using optimal transport, NeurIPS 2019), the soft quantile function is differentiable. We can use gradient descent to update the dynamics model.

>Q5: Discussion of why this approach should be applied to experimental tasks such as "test time flexibility".

A4: Thanks for pointing this out. We will update the writing to add the following explanation. In order to evaluate the ability to generalize to new test-time goals, we run a "test time flexibility" evaluation. "Test time flexibility" is useful to evaluate the capability of the planner as the Diffuser (Janner et al) paper shows.

Long-horizon multi-tasking planning is an important capability of the planner that we need to evaluate experimentally

Offline Reinforcement Learning allows us to evaluate the capacity of our method to recover an effective single-task controller from heterogeneous data of varying quality, which is useful for planning.

Hence, we demonstrate our framework has a number of useful properties and is particularly effective in offline control settings that require long-horizon reasoning and test-time flexibility.

---

### Decision · Program_Chairs · 2023-09-21

**Decision:**

Accept (poster)

**Comment:**

This paper considers the use of conformal prediction framework to explicitly represent uncertainty in learned dynamics model for sequential decision-making. Overall, this is a good direction and the proposed approach is reasonable. The paper is mostly well-written and showed good experimental results.

All reviewers' liked the general approach and agree on its merits. They also raised a number of questions and concerns. Some of them were addressed by authors' rebuttal resulting in score update from some reviewers'.

I recommend accepting the paper because it is a good direction and with the hope that this paper can inspire the NeurIPS community to use conformal prediction based uncertainty quantification for solving sequential decision-making tasks.

I strongly encourage the authors to improve the exposition of the paper and to incorporate the rebuttal discussion/results in the revised paper. Specifically
1. Please clarify the distinction between planning and RL very early in the paper. Clearly mention the paper's scope of contribution as per one of the reviewer comments.
2. Please mention that CP requires calibration data and discuss the practical aspects such as selection and amount from a practitioners' point of view.
3. You may be able to use CP methods to handle distribution shift (see https://arxiv.org/abs/1904.06019 and https://www.stat.berkeley.edu/~ryantibs/statlearn-s23/lectures/conformal_ds.pdf) to demonstrate the applicability of your method to changing rewards to address one of the reviewer comments. It would be great if you can add some results in the final paper. This is important to justify the planning claim of the paper.
4. Please add all the new results and incorporate rebuttal discussion in the final paper.